# Size and depth of monotone neural networks: interpolation and approximation

**Dan Mikulincer**
Massachusetts Institute of Technology

**Daniel Reichman**
Worcester Polytechnic Institute

## Abstract

Monotone functions and data sets arise in a variety of applications. We study the interpolation problem for monotone data sets: The input is a monotone data set with $n$ points, and the goal is to find a size and depth efficient monotone neural network with *non negative parameters* and threshold units that interpolates the data set. We show that there are monotone data sets that cannot be interpolated by a monotone network of depth 2. On the other hand, we prove that for every monotone data set with $n$ points in $\mathbb{R}^d$, there exists an interpolating monotone network of depth 4 and size $O(nd)$. Our interpolation result implies that every monotone function over $[0,1]^d$ can be approximated arbitrarily well by a depth-4 monotone network, improving the previous best-known construction of depth $d+1$. Finally, building on results from Boolean circuit complexity, we show that the inductive bias of having positive parameters can lead to a super-polynomial blow-up in the number of neurons when approximating monotone functions.

## 1 Introduction

The recent successes of neural networks are owed, at least in part, to their great approximation and interpolation power. However, some prediction tasks require their predictors to possess specific properties. This work focuses on monotonicity and studies the effect on overall expressive power when restricting attention to *monotone neural networks*.

Given $x, y \in \mathbb{R}^d$ we consider the partial ordering,

$$x \geq y \iff \text{for every } i = 1, \ldots d, \ [x]_i \geq [y]_i.$$

Here, and throughout the paper, we use $[x]_i$ for the $i^{\text{th}}$ coordinate of $x$. A function $f : [0,1]^d \to \mathbb{R}$ is *monotone*[1] if for every two vectors $x, y \in [0,1]^d$,

$$x \geq y \implies f(x) \geq f(y).$$

Monotone functions arise in several fields such as economics, operations research, statistics, healthcare, and engineering. For example, larger houses typically result in larger prices, and certain features are monotonically related to option pricing [12] and bond rating [9]. As monotonicity constraints abound, there are specialized statistical methods aimed at fitting and modeling monotonic functions such as Isotonic Regression [1, 22, 24] as well as many other works related to monotone approximation [7, 17, 41]. Neural networks are no exception: Several works are devoted to the study of approximating monotone functions using neural networks [9, 35, 38].

When using a network to approximate a monotone function, one might try to "force" the network to be monotone. A natural way to achieve this is to consider only networks where every parameter

---

[1]As we will only be dealing with monotone increasing functions, we shall refer to monotone increasing functions as monotone.

36th Conference on Neural Information Processing Systems (NeurIPS 2022).

(other than the biases) is non-negative[2]. Towards this aim, we introduce the following class of *monotone networks*.

Recall that the building blocks of a neural network are the single one-dimensional neurons $\sigma_{w,b}(x) = \sigma(\langle w, x \rangle + b)$ where $x \in \mathbb{R}^k$ is an input of the neuron, $w \in \mathbb{R}^k$ is a weight parametrizing the neuron, $\sigma : \mathbb{R} \to \mathbb{R}$ is the activation function, and $b \in \mathbb{R}$ is a bias term. Two popular choices for activation functions, that we shall also consider, are the ReLU activation function $\sigma(z) = \max(0, z)$ and the threshold activation function $\sigma(z) = \mathbf{1}(z \geq 0)$, which equals 1 if $z \geq 0$ and zero otherwise. We slightly abuse the term and say that a network is monotone if every single neuron is a monotone function. Since both ReLU and threshold are monotone, this requirement, of having every neuron a monotone function, translates to $w$ having all positive entries.

Such a restriction on the weights can be seen as an inductive bias reflecting prior knowledge that the functions we wish to approximate are monotone. One advantage of having such a "positivity bias" is that it guarantees the monotonicity of the network. Ensuring that a machine learning model approximating a monotone function is indeed monotone is often desirable [15, 25, 26]. Current learning methods such as stochastic gradient descent and back-propagation for training a network are not guaranteed (when applied to a monotone training set) to return a network computing a monotone function. Furthermore, while there are methods for certifying that a given neural network implements a monotone function [25, 38], the task of certifying monotonicity remains a non-trivial task.

Restricting the weights raises several questions on the behavior of monotone neural networks compared to their more general, unconstrained, counterparts. For example, should we still expect monotone networks to be able to approximate arbitrary monotone functions within an arbitrarily small error?

Continuing a line of work on monotone networks, [9, 35], we further elucidate the above comparison and uncover some similarities (a universal approximation theorem) and some surprising differences (the interplay of depth and monotonicity) between monotone and general networks. We will mainly be interested in expressiveness, the ability to approximate monotone functions and to interpolate monotone data sets using monotone neural networks of constant depth.

## 1.1 Our contributions

First, it is not a priori clear that monotone networks can approximate every possible monotone function or even fit any *monotone data set*. Here a monotone data set is a set of $n$ labeled points $(x_i, y_i)_{i \in [n]} \in (\mathbb{R}^d \times \mathbb{R})^n$ with the properties

$$i \neq j \implies x_i \neq x_j \quad \text{and} \quad x_i \leq x_j \implies y_i \leq y_j.$$

In the *monotone interpolation problem* we seek to find a monotone network $N$ such that for every $i \in [n], N(x_i) = y_i$.

**On expressive power and interpolation:** While it is well-known that general networks with ReLU activation are universal approximators (can approximate any continuous function on a bounded domain), perhaps surprisingly, the same is not true for monotone networks and monotone functions. Namely, there are monotone functions that cannot be approximated within an arbitrary small additive error by a monotone network with ReLU gates regardless of the size and depth of the network. This fact was mentioned in [25]: We provide a proof for completeness.

**Lemma 1.** *There exists a monotone function $f : [0, 1] \to \mathbb{R}$ and a constant $c > 0$, such that for any monotone network $N$ with ReLU gates, there exists $x \in [0, 1]$, such that*

$$|N(x) - f(x)| > c.$$

*Proof.* It is known that a sum of convex functions $f_i, \sum \alpha_i f_i$ is convex provided that for every $i, \alpha_i \geq 0$. It is also known that the maximum of convex functions $g_i, \max_i\{g_i\}$ is a convex function. It follows from the definition of the ReLU gate (in particular, ReLU is a convex function) that a neural network with positive weights at all neurons is convex. As there are monotone functions that are not convex, the result follows.

---

[2]We restrict our attention to non negative prediction problems: The domain of the function we seek to approximate does not contain vectors with a negative coordinate.

For a concrete example one may take the function $f(x) = \sqrt{x}$ for which the result holds with $c = \frac{1}{8}$.
$\square$

In light of the above, we shall henceforth only consider monotone networks with a threshold activation function and discuss, for now, the problem of interpolation. For general networks (no restriction on the weights) with threshold activation, it has been established, in the work of Baum [3], that even with 2 layers, for any labeled data set in $\mathbb{R}^d$, there exists an interpolating network.

Our next result is another negative result, which shows that there is an inherent loss of expressive power, when transitioning to 2-layered monotone threshold networks, provided that the dimension is at least two. When the input is real-valued there always exists an interpolating monotone network. This is a simple fact whose proof is omitted: It can be proved using similar ideas to those in [9].

**Lemma 2.** *Let $d \geq 2$. There exists a monotone data set $(x_i, y_i)_{i \in [n]} \in (\mathbb{R}^d \times \mathbb{R})^n$, such that any depth-2 monotone network $N$, with threshold activation must satisfy,*

$$N(x_i) \neq y_i,$$

*for some $i \in [n]$.*

Given the limitation of depth-2 monotone networks, it may seem that, like the case of ReLUs, the class of monotone networks with threshold activation is too limited, in the sense that it cannot approximate any monotone function, with constant depth (allowing the depth to scale with the input dimension was considered in [9], see below). One reason for such a belief is that, for non-monotone networks, depth 2 suffices to ensure universality: any continuous function over a bounded domain can be approximated by a depth-2 network [2, 8, 19] and this universality result holds for networks with threshold or ReLUs as activation functions. Our first main result supports the contrary to this belief. We establish a depth separation result for monotone threshold networks, and show that, by slightly increasing the number of layers, monotone networks can interpolate arbitrary monotone data sets. Thereafter a simple argument shows that monotone networks of bounded depth become universal approximators of monotone functions. As noted, this is in sharp contrast to general neural networks, where adding extra layers can affect the efficiency of the representation [13], but does not change the expressive power.

**Theorem 1.** *Let $(x_i, y_i)_{i \in [n]} \in (\mathbb{R}^d \times \mathbb{R})^n$ be a monotone data set. There exists a monotone threshold network $N$, with 4 layers and $O(nd)$ neurons such that,*

$$N(x_i) = y_i,$$

*for every $i \in [n]$.*

*Moreover, if the set $(x_i)_{i \in [n]}$ is totally-ordered, in the sense that, for every $i, j \in [n]$, either $x_i \leq x_j$ or $x_j \geq x_i$, then one may take $N$ to have 3 layers and $O(n)$ neurons.*

We also complement Theorem 1 with a lower bound that shows that the number of neurons we use is essentially tight, up to the dependence on the dimension.

**Lemma 3.** *There exists a monotone data set $(x_i, y_i)_{i \in [n]} \subset (\mathbb{R}^d \times \mathbb{R})^n$ such that, if $N$ is an interpolating monotone threshold network, the first layer of $N$ must contain $n$ units. Moreover, this lower bound holds when the set $(x_i)_{i \in [n]}$ is totally-ordered.*

The lower bound of Lemma 3 demonstrates another important distinction between monotone and general neural networks. According to [3], higher-dimensions allow general networks, with 2 layers, to be more compact. Since the number of parameters in the networks increase with the dimension, one can interpolate labeled data sets in general position with only $O\left(\frac{n}{d}\right)$ neurons. Moreover, for deeper networks, a recent line of work, initiated in [40], shows that $O(\sqrt{n})$ neurons suffice. Lemma 3 shows that monotone networks cannot enjoy the same speed-up, either dimensional or from depth, in efficiency.

Since we are dealing with monotone functions, our interpolation results immediately imply a universal approximation theorem for monotone networks of depth 4.

**Theorem 2.** *Let $f : [0, 1]^d \to \mathbb{R}$ be a continuous monotone function and let $\varepsilon > 0$. Then, there exists a monotone threshold network $N$, with 4 layers, such that, for every $x \in [0, 1]^d$,*

$$|N(x) - f(x)| \leq \varepsilon.$$

*If the function $f$ is L-Lipschitz, for some $L > 0$, one can take $N$ to have $O\left(d\left(\frac{L\sqrt{d}}{\varepsilon}\right)^d\right)$ neurons.*

While it was previously proven, in [9], that monotone networks with threshold activation can approximate any monotone function, the *depth* in the approximating network given by [9] scales *linearly* with the input dimension. Our result is thus a significant improvement which only requires constant depth. When looking at the size of the network, the linear depth construction, in [9], iteratively evaluates Riemann sums, and builds a network which is piecewise-constant on a grid. Hence, for $L$-Lipschitz functions, it would require a similar amount of neurons. Again, we see that comparable results can be obtained while maintaining constant depth. Whether one can achieve similar results to Theorems 1 and 2 with only 3 layers is an interesting question which we leave for future research.

**Efficiency when compared to general networks:** We have shown that, with 4 layers, monotone networks can serve as universal approximates. However, even if a monotone network can approximate a monotone function arbitrarily well it might be that it requires a much larger size, when compared to unconstrained networks. In this case, the cost of having a much larger network might outweigh the benefit of having a network that is guaranteed to be monotone.

In our second main result we verify that this can sometimes be the case. We show that using monotone networks to approximate, in the $\ell_\infty$-norm, a monotone function $h : [0,1]^d \to \mathbb{R}$ can lead to a super-polynomial blow-up in the number of neurons. Namely, we demonstrate a smooth monotone function $h : [0,1]^d \to \mathbb{R}$ with a poly$(d)$ Lipschitz constant such that $h$ can be approximated within an additive error of $\varepsilon > 0$ by a general neural network with poly$(d)$ neurons. Yet any *monotone* network approximating $h$ within error smaller than $\frac{1}{2}$ requires super-polynomial size in $d$.

**Theorem 3.** *There exists a monotone function $h : [0,1]^d \to \mathbb{R}$, such that:*

- *Any monotone threshold network $N$ which satisfies,*

$$|N(x) - h(x)| < \frac{1}{2}, \text{ for every } x \in [0,1]^d,$$

  *must have $e^{\Omega(\log^2 d)}$ neurons.*

- *For every $\varepsilon > 0$, there exists a general threshold network $N$, which has $\mathrm{poly}(d)$ neurons and such that,*
$$|N(x) - h(x)| < \varepsilon, \text{ for every } x \in [0,1]^d.$$

## 2 Related work

We are not aware of previous work studying the interpolation problem for monotone data sets using monotone networks. For general data sets and interpolation using networks with no positivity requirement there is extensive research regarding the size and depth needed to achieve interpolation [5, 10, 40, 44] starting with the seminal work of Baum studying this problem for networks with threshold units [3]. Known constructions of neural networks achieving interpolations are non-monotone: they may result in negative parameters even for monotone data sets.

Several works have studied approximating monotone (real) functions over a bounded domain using a monotone network. Sill [35] provides a construction of a monotone network (all parameters are non-negative) with depth 3 where the first layer consists of linear units divided into groups, the second layer consists of max gates where each group of linear units of the first layer is fed to a different gate and a final gate computing the minimum of all outputs from the second layer. It is proven in [35] that this class of networks can approximate every monotone function over $[0,1]^d$. We remark that this is very different than the setting considered in this present work. First, using both min and max gates in the same architecture with positive parameters does not fall into the modern paradigm of an activation function. Moreover, we are not aware of works prior to Theorem 2, that show how to implement or approximate min and max gates, with arbitrary fan-ins, using constant depth *monotone networks*. [3] Finally, the results from [35] focus on approximating arbitrary monotone functions and do not consider the monotone interpolation problem studied here.

---

[3]There are constructions of depth-3 threshold circuits with discrete inputs that are given $m$ numbers each represented by $n$ bits and compute the maximum of these numbers [37] This setting is different from ours where the inputs are real numbers.

Later, the problem of approximating arbitrary monotone functions with networks having non-negative parameters using more standard activation functions such as thresholds or sigmoids has been studied in [9]. In particular [9] gives a recursive construction showing how to approximate in the $\ell_\infty$ norm an arbitrary monotone function using a network of depth $d + 1$ ($d$-hidden layers) with threshold units and non-negative parameters. In addition [9] provides a construction of a monotone function $g : [0, 1]^2 \to \mathbb{R}$ that cannot be approximated in the $\ell_\infty$ norm with an error smaller than $1/8$ by a network of depth 2 with sigmoid activation and non-negative parameters, regardless of the number of neurons in the network. Our Lemma 2 concerns networks with threshold gates and applies to arbitrary dimension larger than 1. It can also be extended to provide monotone functions that cannot be approximated by monotone networks with thresholds of depth 2.

Lower bounds for monotone models of computation have been proven for a variety of models [11], including monotone De Morgan[4] circuits [14, 18, 30, 31], monotone arithmetic circuits and computations [6, 21, 43], which correspond to polynomials with non-negative coefficients, and circuits with monotone real gates [20, 28] whose inputs and outputs are *Boolean*. One difference between our separation result regarding monotone and non-monotone networks and these works is that we consider *real* differentiable functions as opposed to Boolean functions. Furthermore we need functions that can be computed by an unconstrained neural network of *polynomial size*. In contrast, known lower bounds for circuits with real monotone gates apply to Boolean functions that are believed to be intractable (require a super polynomial number of gates) to compute even with non-monotone circuits (e.g., deciding if a graph contains a clique of a given size). Finally our model of computation of neural networks with threshold gates differs from arithmetic circuits [34] which use gates that compute polynomial functions.

To achieve our separation result we begin with a Boolean function $m$, which requires a super-polynomial size to compute by any Boolean circuit with monotone threshold gates, but can be computed efficiently with arbitrary threshold circuits: the existence of $m$ follows from [31]. Thereafter we show how to smoothly extend $m$ to have domain $[0, 1]^d$ while preserving monotonicity. Proving lower bounds for neural networks with a continuous domain by extending a Boolean function $f$ for which lower bounds are known to a function $f'$ whose domain is $[0, 1]^d$ has been done before [39, 42]. However the extension method in these works do not yield a function that is monotone. Therefore, we use a different method based on the multi-linear extension.

## 3 Preliminaries and notation

We work on $\mathbb{R}^d$, with the Euclidean inner product $\langle \cdot, \cdot \rangle$. For $k \in \mathbb{N}$, we denote $[k] = \{1, 2, \ldots, k\}$ and use $\{e_i\}_{i \in [d]}$ for standard unit vectors in $\mathbb{R}^d$. That is, for $i \in [d]$, $e_i = (\underbrace{0, \ldots, 0}_{i-1 \text{ times}}, 1, \underbrace{0, \ldots, 0}_{d-i \text{ times}})$.

For $x \in \mathbb{R}^d$ and $i \in [d]$, we write $[x]_i := \langle x, e_i \rangle$, the $i^{\text{th}}$ coordinate of $x$.

With a slight abuse of notation, when the dimension of the input changes to, say, $\mathbb{R}^k$, we will also use $\{e_i\}_{i \in [k]}$ to stand for standard unit vectors in $\mathbb{R}^k$. To avoid confusion, we will always make sure to make the dimension explicit.

A neural network of depth $L$ is a function $N : \mathbb{R}^d \to \mathbb{R}$, which can be written as a composition,

$$N(x) = N_L(N_{L-1}(\ldots N_2(N_1(x)) \ldots),$$

where for $\ell \in [L]$, $N_\ell : \mathbb{R}^{d_\ell} \to \mathbb{R}^{d_{\ell+1}}$ is a layer. We set $d_1 = d$, $d_{L+1} = 1$ and term $d_{\ell+1}$ as the width of layer $\ell$. Each layer is composed of single neurons in the following way: for $i \in [d_{\ell+1}]$, $[N_\ell(x)]_i = \sigma(\langle w_i^\ell, x \rangle + b_i^\ell)$ where $\sigma : \mathbb{R} \to \mathbb{R}$ is the activation function, $w_i^\ell \in \mathbb{R}^{d_\ell}$ is the weight vector, and $b_i^\ell \in \mathbb{R}$ is the bias. The only exception is the last layer which is an affine functional of the previous layers, $N_L(x) = \langle w^L, x \rangle + b^L$. For a weight vector $w^L \in \mathbb{R}^{d_L}$ and bias $b^L \in \mathbb{R}$.

Suppose that the activation function is monotone. We say that a network $N$ is monotone, if, for every $\ell \in [L]$ and $i \in [d_{\ell+1}]$, the weights vector $w_i^\ell$ has all positive coordinates. In other words, $[w_i^\ell]_j \geq 0$, for every $j \in [d_\ell]$.

---

[4]Circuits with AND as well as OR gates without negations.

## 4 A counter-example to expressibility

We outline the ideas behind Lemma 2. We find it instructive to consider the case $d = 2$. Lemma 2 follows via a simple generalization of this case: a proof[5] of the Lemma can be found in the supplementary material. Recall that $\sigma(x) = \mathbf{1}(x \geq 0)$ is the threshold function and consider the monotone set,

$$
\begin{aligned}
x_1 &= (2, 0), \ y_1 = 0 \\
x_2 &= (0, 2), \ y_2 = 0 \\
x_3 &= (1, 1), \ y_3 = 1.
\end{aligned}
$$

Assume towards a contradiction that there exists a network $N(x) := \sum_{m=1}^{r} a_m \sigma \left( \langle x, w_m \rangle - b_m \right)$, which interpolates the set. Set

$$
I = \{ m \in [r] | \langle x_3, w_m \rangle \geq b_m \} = \{ m \in [r] | [w_m]_1 + [w_m]_2 \geq b_m l \}.
$$

The set $I$ is the set of all neurons which are active on $x_3$, and, since $N(x_3) = 1$, $I$ is non-empty. We also define,

$$
\begin{aligned}
I_1 &= \{ m \in I | [w_m]_1 \geq [w_m]_2 \} \\
I_2 &= \{ m \in I | [w_m]_2 \geq [w_m]_1 \}.
\end{aligned}
$$

It is clear that $I_1 \cup I_2 = I$. Observe that for $m \in I_1$, by monotonicity, we have $\langle x_1, w_m \rangle = 2[w_m]_1 \geq [w_m]_1 + [w_m]_2 = \langle x_3, w_m \rangle$. Since the same also holds for $m \in I_2$ and $x_2$, we have,

$$
\begin{aligned}
N(x_1) + N(x_2) &\geq \sum_{m \in I_1} a_m \sigma \left( \langle x_1, w_m \rangle - b_m \right) + \sum_{m \in I_2} a_m \sigma \left( \langle x_2, w_m \rangle - b_m \right) \\
&\geq \sum_{m \in I_1} a_m \sigma \left( \langle x_3, w_m \rangle - b_m \right) + \sum_{m \in I_2} a_m \sigma \left( \langle x_3, w_m \rangle - b_m \right) \\
&\geq \sum_{m \in I} a_m \sigma \left( \langle x_3, w_m \rangle - b_m \right) = N(x_1) = 1.
\end{aligned}
\tag{1}
$$

Hence, either $N(x_1) \geq \frac{1}{2}$ or $N(x_2) \geq \frac{1}{2}$.

## 5 Four layers suffice with threshold activation

Let $(x_i, y_i)_{i=1}^{n} \in (\mathbb{R}^d \times \mathbb{R})^n$ be a monotone data set, and assume, with no loss of generality,

$$
0 \leq y_1 \leq y_2 \leq \cdots \leq y_n.
\tag{2}
$$

If, for some $i, i' \in [n]$ with $i \neq i'$ we have $y_i = y_{i'}$, and $x_i \leq x_{i'}$, we will assume $i < i'$. Other ties are resolved arbitrarily. Note that the assumption that the $y_i$'s are positive holds without loss of generality, as one can always add a constant to the output of the network to handle negative labels.

This section is dedicated to the proof of Theorem 1, and we will show that one can interpolate the above set using a monotone network with 3 hidden layers. The first hidden layer is of width $dn$ and the second and third of width $n$.

Throughout we shall use $\sigma(t) = \mathbf{1}(t \geq 0)$, for the threshold function. For $\ell \in \{1, 2, 3\}$ we will also write $(w_i^\ell, b_i^\ell)$ for the weights of the $i^{\text{th}}$ neuron in level $\ell$. We shall also use the shorthand, $\sigma_i^\ell(x) = \sigma(\langle x, w_i^\ell \rangle - b_i^\ell)$.

We first describe the first two layers. The second layer serves as a monotone embedding into $\mathbb{R}^n$. We emphasize this fact by denoting the second layer as $E : \mathbb{R}^d \to \mathbb{R}^n$, with $i^{\text{th}}$ coordinate is given by, $[E(x)]_i = \sigma_i^2(N_1(x))$, where $[N_1(x)]_j = \sigma_j^1(x)$, for $j = 1, \ldots, nd$, are the outputs of the first layer.

---

[5]The 2D example here as well its generalization for higher dimensions can be easily adapted to give an example of a monotone function that cannot be approximated in $\ell_2$ by a depth-two monotone threshold network.

**First hidden layer:** The first hidden layer has $dn$ units. For $j = 1, \ldots, dn$. We let $e_i$ be the $i^{\text{th}}$ standard basis vector in $\mathbb{R}^d$ and define,

$$\sigma_j^1(x) := \sigma\left(\langle x, e_{(j \bmod d)+1}\rangle - \langle x_{\lceil \frac{j}{d}\rceil}, e_{(j \bmod d)+1}\rangle\right).$$

In other words, $w_j^1 = e_{(j \bmod d)+1}$ and $b_j^1 = \langle x_{\lceil \frac{j}{d}\rceil}, e_{(j \bmod d)+1}\rangle$ (the addition of 1 offsets the fact that mod operations can result in 0). To get a feeling of what the layer does, suppose that $j \equiv r \bmod d$, then unit $j$ is activated on input $x$ iff the $(r+1)^{\text{th}}$ entry of $x$ is at least the $(r+1)^{\text{th}}$ entry of $x_{\lceil \frac{j}{d}\rceil}$.

**Second hidden layer:** The second layer has $n$ units. For $j = 1, \ldots, nd$, with a slight abuse of notation we now use $e_j$ for the $j^{\text{th}}$ standard basis vector in $\mathbb{R}^{nd}$ and define unit $i = 1, \ldots, n$, $\sigma_i^2 : \mathbb{R}^{nd} \to \mathbb{R}$, by,

$$\sigma_i^2(y) = \sigma\left(\sum_{r=1}^d \langle y, e_{d(i-1)+r}\rangle - d\right).$$

Explicitly, $w_i^2 = \sum_{r=1}^d e_{d(i-1)+r}$ and $b_i^2 = d$. With this construction in hand, the following is the main property of the first two layers.

**Lemma 4.** *Let $i = 1, \ldots, n$. Then, $[E(x)]_i = 1$ if and only if $x \geq x_i$. Otherwise, $[E(x)]_i = 0$.*

*Proof.* By construction, we have $[E(x)]_i = 1$ if and only if $\sum_{r=1}^d \sigma_{d(i-1)+r}^1(x) \geq d$. For each $r \in [d]$, $\sigma_{d(i-1)+r}^1(x) \in \{0, 1\}$. Thus, $[E(x)]_i = 1$ if and only if, for every $r \in [d]$, $\sigma_{d(i-1)+r}^1(x) = 1$. But $\sigma_{d(i-1)+r}^1(x) = 1$ is equivalent to $[x]_r \geq [x_i]_r$. Since this must hold for every $r \in [d]$, we conclude $x \geq x_i$. □

The following corollary is now immediate.

**Corollary 4.** *Fix $j \in [n]$ and let $i \in [n]$. If $j < i$, then $[E(x_j)]_i = 0$. If $j \geq i$, then there exists $i' \geq i$ such that $[E(x_j)]_{i'} = 1$.*

*Proof.* For the first item, if $j < i$, by the ordering of the labels (2), we know that $x_j \not\geq x_i$. By Lemma 4, $[E(x_j)]_i = 0$.

For the second item, by construction, $[E(x_j)]_j = 1$. Since $j \geq i$, the claim concludes. □

**The third hidden layer:** The third layer contains $n$ units with weights given by $w_i^3 = \sum_{r=i}^n e_r$ and $b_i^3 = 1$. Thus,

$$\sigma_i^3(E(x)) = \sigma\left(\sum_{r=i}^n [E(x)]_r - 1\right). \tag{3}$$

**Lemma 5.** *Fix $j \in [n]$ and let $i \in [n]$. $\sigma_i^3(E(x_j)) = 1$ if $j \geq i$ and $\sigma_i^3(E(x_j)) = 0$ otherwise.*

*Proof.* By Corollary 4, $[E(x_j)]_r = 0$, for every $r > j$. In particular, if $i > j$, by (3), we get, $\sigma_i^3(E(x_j)) = \sigma(-1) = 0$. On the other hand, if $j \geq i$, then by Corollary 4, there exists $i' \geq i$ such that $[E(x_j)]_{i'} = 1$, and, $\sigma_i^3(E(x_j)) = \sigma([E(x_j)]_{i'} - 1) = 1$. □

**The final layer:** The fourth and final layer is a linear functional of the output of the third layer. Formally, for $x \in \mathbb{R}^d$, the output of the network is, $N(x) = \sum_{i=1}^n [w^4]_i \sigma_i^3(E(x))$, for some weights vector $w^4 \in \mathbb{R}$. To complete the construction we now define the entries of $w^4$, as $[w^4]_i = y_i - y_{i-1}$ with $y_0 = 0$. We are now ready to prove Theorem 1.

*Proof of Theorem 1.* Consider the function $N(x) = \sum_{i=1}^{n} [w^4]_i \sigma_i^3(E(x))$ described above. Clearly, it is a network with 3 hidden layers, by construction. To see that it is monotone, observe that for $\ell \in \{1, 2, 3\}$ each $w_i^\ell$ is a sum of standard basis vectors, and thus has non-negative entries. The weight vector $w^4$ also has non-negative entries, since, by assumption, $y_i \geq y_{i-1}$.

We now show that $N$ interpolates the data set $(x_j, y_j)_{j=1}^n$. Indeed, fix $j \in [n]$. By Lemma 5, we have, $N(x_j) = \sum_{i=1}^{n} [w^4]_i \sigma_i^3(E(x_j)) = \sum_{i=1}^{j} [w^4]_i = \sum_{i=1}^{j} (y_i - y_{i-1}) = y_j - y_0 = y_j$. The proof is complete, for the general case.

To handle the case of totally-ordered $(x_i)_{i \in [n]}$, we slightly alter the construction of the first two layers, and compress them into a single layer satisfying Lemma 4, and hence Lemma 5.

The total-order of $(x_i)_{i \in [n]}$ implies the following: For every $i \in [n]$, there exists $r(i) \in [d]$, such that for any $j \in [n]$, $[x_i]_{r(i)} < [x_j]_{r(i)}$ if and only if $i < j$. In words, for every point in the set, there exists a coordinate which separates it from all the smaller points. We thus define $w_i^1 = e_{r(i)}$ and $b_i = 1$.

From the above it is clear that $\sigma_i^1(x_j) = \sigma([x_j]_{r(i)} - 1) = \begin{cases} 1 & \text{if } i \leq j \\ 0 & \text{if } i > j \end{cases}$.

As in the general case, we define $E(x) : \mathbb{R}^d \to \mathbb{R}^n$ by $[E(x)]_i = \sigma_i^1(x)$, and note that Lemma 4 holds for this construction of $E$ as well. The next two layers are constructed exactly like in the general case and the same proof holds. □

We conclude this section with two comments. First, as noted, our interpolation scheme can be used to prove that any continuous monotone function can be approximated by a monotone threshold network which is Theorem 2. A proof can be found in the supplementary. Second, interpolating monotone networks are wide: It turns out, $\Omega(n)$ neurons may be needed for monotone interpolation. A proof of this (Lemma 3) can be found in the supplementary as well. For general threshold networks (no restriction on the weights) one can interpolates $n$ points using $O(\sqrt{n} + f(\delta))$ neurons where $f$ depends on the minimal distance between any two of the data points [29, 40]. Hence Lemma 3 shows that there exist monotone data sets such that interpolating them with a monotone network entails a quadratic blowup in the size of the network even if the data set is well separated.

## 6 A super polynomial separation between the size of monotone and arbitrary threshold networks

By the universal approximation result for monotone threshold networks, Theorem 2, we can approximate monotone functions by monotone networks. Are there functions such that monotone networks approximating them provably require a much larger size than networks which are allowed to have negative parameters as well? We show that the answer is positive when seeking an $\ell_\infty$-approximation smaller than $\varepsilon$ for any $\varepsilon \in [0, 1/2)$.

Our proof of this fact builds on findings from monotone complexity theory of Boolean functions. Given an undirected graph $G = (V, E)$ with $2n$ vertices a *matching* $M$ is a set of pairwise disjoint edges. A *perfect* matching is a matching of size $n$ (which is largest possible). There are efficient algorithms for deciding if a bipartite graph has a perfect matching [23]. Furthermore, by standard results that convert Turing machines that decide an algorithmic problem with inputs of size $n$ in time $t(n)$ to threshold circuits with $O(t(n)^2)$ gates, [32, 36], it follows that there is a network of size polynomial in $n$ that decides, given the incidence matrix of a graph, whether it has a perfect matching. A seminal result by Rzaborov [31] shows that the *monotone* complexity of the matching function is not polynomial.

**Theorem 5.** *Let $g$ be the Boolean function that receives the adjacency matrix of a $2n$-vertex graph $G$ and decides if $G$ has a perfect matching. Then, any Boolean circuit with AND and OR gates that computes $g$ has size $n^{\Omega(\log n)}$. Furthermore, the same lower bound applies if the graph is restricted to be a bipartite graph $G(A, B, E)$ where $|A| = |B| = n$ is the bi-partition of the graph.*

**Definition 6** (Matching probabilities in non-homogeneous random bipartite graphs). Let $\mathbf{p} = (p_{ij})_{i,j=1}^n \in [0, 1]^{n \times n}$ and define $G(\mathbf{p})$ to be a random bipartite graph on vertex set $[n] \times [n]$,

such that each edge $(i, j)$ appears independently with probability $p_{ij}$, for every $1 \leq i, j \leq n$. Define $m : [0, 1]^{n \times n} \to [0, 1]$ as[6], $m(\mathbf{p}) = \mathbb{P}\left(G(\mathbf{p}) \text{ contains a perfect matching}\right)$.

When $\mathbf{p} \in \{0, 1\}^{n \times n}$, $m(\mathbf{p})$ reduces to the indicator function of a perfect matching in a given bipartite graph. Thus $m(\mathbf{p})$ should be thought of as the harmonic (or multi-linear) extension of the indicator function to the solid cube $[0, 1]^{n \times n}$.

Theorem 3 is an immediate consequence of the following more specific theorem, which is our main result for this section.

**Theorem 7.** *The function $m$ defined above is a smooth monotone function with Lipschitz constant $\leq n$, which satisfies the following: If $N$ is a monotone threshold network of size $e^{o((\log n)^2)}$, there exists $\mathbf{p} \in [0, 1]^{n \times n}$, such that, $|N(\mathbf{p}) - m(\mathbf{p})| \geq \frac{1}{2}$. Furthermore, for every fixed $\varepsilon > 0$ there exist a general threshold network $N$ of polynomial size in $n$, such that for all $\mathbf{p} \in [0, 1]^{n \times n}$, $|N(\mathbf{p}) - m(\mathbf{p})| \leq \varepsilon$.*

Our proof of Theorem 7 is divided into three parts. We first establish the relevant properties of $m$. We then show that $m$ cannot be approximated by a monotone network of polynomial size. Finally, we show that $m$ can be approximated, arbitrarily well, by a general network with polynomial size. We begin by collecting several simple facts about the function $m$. All missing proofs can be found in the supplementary. A standard coupling argument (omitted) shows that $m$ is monotone.

We have that $m$ is Lipschitz continuous: Let $\mathbf{p}, \mathbf{p}' \in [0, 1]^{n \times n}$. Then, $|m(\mathbf{p}) - m(\mathbf{p}')| \leq n\|\mathbf{p} - \mathbf{p}'\|$. This is a simple consequence of the fact that $m$ is the harmonic extension of a bounded function. The fact that $m$ is smooth can again be seen from the multi-linear extension. Proofs that $m$ has these properties can be found in the supplementary. The proof (whose details can be found in the supplementary) that the function $m$ cannot be approximated by a monotone network of polynomial size uses the fact [4] that a threshold gate with $s$ inputs, non negative parameters and Boolean inputs can be computed by a monotone De Morgan circuit of size poly$(s)$. Therefore, if there was a monotone network of size $e^{o((\log n)^2)}$ approximating $m(x)$ we could replace each gate with a monotone De Morgan circuit entailing a polynomial blowup to the size of the network. This in turn would imply the existence of a monotone De Morgan circuit of size $e^{o((\log n)^2)}$ computing $m$ over Boolean inputs which would contradict[7] Theorem 5. Summarizing:

**Lemma 6.** *If $N$ is a monotone threshold network of size $e^{o((\log n)^2)}$, there exists $\mathbf{p} \in [0, 1]^{n \times n}$, such that, $|N(\mathbf{p}) - m(\mathbf{p})| \geq \frac{1}{2}$.*

Finally, we show how to approximate $m$ with a network (without weight restrictions) of polynomial size. To estimate the probability a graph $G(\mathbf{p})$ drawn according to a probability vector $\mathbf{p}$ has a perfect matching we can realize independently (polynomially) many copies of graphs distributed as $G(\mathbf{p})$ and estimate the number of times a perfect matching is detected. In order to implement this idea two issues need to be addressed: the use of randomness by the algorithm (our neural networks do not use randomness) and the algorithm dealing with probability vectors in $[0, 1]^{n \times n}$ that may need infinitely many bits to be represented in binary expansion.

We first present a randomized polynomial-time algorithm, denoted $A$, for approximating $m(\mathbf{p})$. We then show how to implement it with a (deterministic) threshold network. Algorithm $A$ works as follows. Let $q(), r()$ be polynomials to be defined later. First, the algorithm (with input $\mathbf{p}$) only considers the $q(n)$ most significant bits in the binary representation of every coordinate in $\mathbf{p}$. Next it realizes $r(n)$ independent copies of $G(\mathbf{p})$. It checks[8] for each one of these copies whether it contains a perfect matching of size $n$. Let $t$ be the number of times a perfect matching is detected ($t$ depends on $\mathbf{p}$: We omit this dependency to lighten up notation). The algorithm outputs $A(\mathbf{p}) := \frac{t}{r(n)}$. Clearly the running time of this algorithm is polynomial.

Let $\widetilde{\mathbf{p}}$ be the vector obtained from $\mathbf{p}$ when considering the $q(n)$ most significant bits in each coordinate, and observe $A(\mathbf{p}) \stackrel{\text{law}}{=} A(\widetilde{\mathbf{p}})$. We next show how to implement this algorithm by a neural network of polynomial size that does not use randomness. For a proof please see the supplementary.

---

[6]The function $m$ depends also on $n$ but we omit this dependency as it is always clear from the context.

[7]This shows that the size of such a monotone network cannot be polynomial in $n$.

[8]We use the flow-based poly-time algorithm to decide if a bipartite graph has a perfect matching.

**Lemma 7.** *Let $\delta \in (0,1)$ be a fixed constant. Then, if $\tilde{p} \in [0,1]^{n \times n}$ is such that every coordinate $\tilde{\mathbf{p}}_{\mathbf{ij}}$ can be represented by at most $q(n)$ bits there exists a neural network of polynomial size $N$ such that for every $\mathbf{p} \in [0,1]^{n \times n}, |m(\mathbf{p}) - N(\mathbf{p}))| \leq \delta + \frac{n^2}{\sqrt{2}^{q(n)}}$.*

We can now prove the following Lemma concluding the proof of Theorem 7.

**Lemma 8.** *For every fixed $\varepsilon > 0$ there exist a general threshold network $N$ of polynomial size in $n$, such that for all $\mathbf{p} \in [0,1]^{n \times n}$, $|N(\mathbf{p}) - m(\mathbf{p})| \leq \varepsilon$.*

*Proof.* Set $\delta = \frac{\varepsilon}{2}$ and let $N$ be the network constructed in Lemma 7 with accuracy parameter $\delta$. Choose $q(n)$ which satisfies $q(n) > \log(\frac{4n^4}{\varepsilon^2})$. Thus, Lemma 7 implies, for every $\mathbf{p} \in [0,1]^{n \times n}$:

$$|m(\mathbf{p}) - N(\mathbf{p}))| \leq \delta + \frac{n^2}{\sqrt{2}^{q(n)}} \leq \frac{\varepsilon}{2} + \frac{\varepsilon}{2} = \varepsilon.$$

$\square$

# 7  Conclusion

We studied neural networks with nonnegative weights and examined their power and limitation in approximating monotone functions.

Our results reveal that for the ReLU activation, restricting the weights to be nonnegative severally limits the ability of the model to express monotone functions. For threshold activation we have shown that the restriction to positive parameters is less severe and that universality can be achieved at constant depth. In addition, we have shown that monotone neural networks can be much more resource consuming, in terms of the number of neurons needed to approximate monotone functions.

We focused on the threshold activation function. It is an interesting direction to extend our results for other activation functions such as sigmoids. For the universality result of depth 4 monotone function it seems plausible one could approximate thresholds by sigmoids to prove that monotone networks of depth 4 with sigmoids are universal approximators of monotone functions. For our lower bounds based on the matching function $m$ it appears that new ideas are needed to show a super polynomial separation between the size needed for monotone as opposed to arbitrary networks with sigmoids to approximate $m$.

The field of Boolean circuit complexity has been used before to address theoretical questions related to neural networks (e.g., [6, 16, 27, 33]) and we believe that additional insights could be found by studying the intersection of circuit complexity and deep learning. With regard to monotone neural networks it is likely that stronger lower bounds can be proved when the depth of the network is bounded. Proving separation between monotone and non monotone networks with respect to the square loss is another avenue for further research.

One aspect we did not consider here is learning neural networks with positive parameters using gradient descent. It would be interesting to examine the efficacy of gradient methods both empirically and theoretically. Such study could lead to further insights regarding methods that ensure that a neural network approximating a monotone function is indeed monotone. Finally, we did not deal with generalization properties of monotone networks: Devising tight generalization bounds for monotone networks is left for future study.

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
