## A   Omitted proofs

### A.1   Counter-example to expressibility when $d \geq 2$

*Proof of Lemma 2.*  Consider the following monotone set of $d+1$ data points in $\mathbb{R}^d$ with $d \geq 2$. For $i \in [d], x_i = d \cdot e_i$, is a vector whose $i^{\text{th}}$ coordinate is $d$ and all other coordinates are $0$, and set $y_i = 0$. We further set $x_{d+1} = (1, \ldots, 1)$ (the all 1's vector) with $y_{d+1} = 1$.

Suppose towards a contradiction there is a monotone depth-2 threshold network

$$N(x) = \sum_{m=1}^{r} a_m \sigma(\langle x, w_m \rangle - b_m),$$

with $N(x_{d+1}) = 1$ and for every $i \in [d], N(x_i) = 0$. We prove the result while assuming that the bias of the output layer is $0$. Since the bias just adds a constant to every output it is straightforward to take it into account.

Denote,

$$I := \left\{ m \in [r] \mid \sum_{i=1}^{d} [w_m]_i \geq b_m \right\}.$$

Since $N(x_{d+1}) = 1$, we have that $I$ is non-empty. For $j \in [d]$, let

$$I_j := \{ m \in I \mid [w_m]_j = \max\{[w_m]_1, \ldots, [w_m]_d\}\}.$$

Clearly $I = \bigcup_{j=1}^{d} I_j$ and we can assume, with no loss of generality, that this is a disjoint union. Now, by following the exact same logic as in (1),

$$\sum_{i=1}^{d} N(x_i) \geq \sum_{m \in I_1} a_m \sigma(\langle x_1, w_m \rangle - b_m) + \ldots + \sum_{m \in I_d} a_m \sigma(\langle x_d, w_m \rangle - b_m)$$

$$\geq \sum_{m \in I} a_m \sigma(\langle x_{d+1}, w_m \rangle - b_m) = 1$$

Therefore there exists $j \in [d]$ with $N(x_j) \geq \frac{1}{d} > 0$, which is a contradiction. $\qquad\square$

## A.2 Lower bounds

*Proof of Lemma 3.* Let $(x_i, y_i)_{i \in [n]} \in (\mathbb{R}^d \times \mathbb{R})^n$ be a monotone data set, such that for every $1 \leq i \leq n - 1$, $x_i \leq x_{i+1}$ and $y_i \neq y_{i+1}$. Suppose that the first layer of $N$ has $k$ units and denote it by $N_1 : \mathbb{R}^d \to \mathbb{R}^k$. Now, let $1 \leq i < j \leq n$. Since, every unit is monotone, and since $x_i \leq x_j$, we have for $\ell \in [k]$ the implication

$$[N_1(x_i)]_\ell \neq 0 \implies [N_1(x_i)]_\ell = [N_1(x_j)]_\ell.$$

Denote $I_i = \{\ell \in [k] | [N_1(x_i)]_\ell \neq 0\}$. The above shows the following chain condition,

$$I_1 \subset I_2 \subset \cdots \subset I_n.$$

Suppose that $k < n$, then as $\{I_i\}_{i=1}^{n}$ is an ascending chain of subsets in $[k]$, necessarily there exists an $i < n$, such that $I_i = I_{i+1}$, which implies $N_1(x_i) = N_1(x_{i+1})$. We conclude that $N(x_i) = N(x_{i+1})$, which cannot happen as $y_i \neq y_{i+1}$. Hence, necessarily $k > n$. $\qquad\square$

## A.3 Universal approximation

*Proof of Theorem 2.* Since $f$ is continuous and $[0, 1]^d$ is compact, $f$ is uniformly continuous. Hence, there exists $\delta > 0$, such that

$$\|x - y\| \leq \delta \implies |f(x) - f(y)| \leq \varepsilon, \text{ for all } x, y \in [0, 1]^d.$$

Set $\delta_d = \frac{\delta}{\sqrt{d}}$ and consider the grid,

$$G_\delta = (\delta_d \mathbb{Z})^d \cap [0, 1]^d.$$

I.e. $G_\delta$ is a uniform grid of points with spaces of width $\delta_d$. Define now a monotone data set $(x_j, f(x_j))_{x_j \in G_\delta}$. By Theorem 1 there exists a network $N$, such that $N(x_j) = f(x_j)$ for every $x_j \in G_\delta$. We claim that this is the approximating network. Let $x \in [0, 1]^d$ and let $x_-$ (resp. $x_+$) be the closest point to $x$ in $G_\delta$ such that $x \geq x_-$ (resp. $x \leq x_+$). Observe that, $x_-$ and $x_+$ are vertices of a sub-cube in $[0, 1]^d$ of side length $\delta_d$. Thus, $\|x_- - x_+\| \leq \sqrt{d} \delta_d = \delta$. Now, since both $N$ and $f$ are monotone,

$$|N(x) - f(x)| \leq \max\left(|N(x_+) - f(x_-)|, |f(x_+) - N(x_-)|\right) = |f(x_+) - f(x_-)| \leq \varepsilon.$$

Assume now that $f$ is $L$-Lipschitz and Let us now estimate the size of the network. According to Theorem 1, the network has $O(d|G_\delta|)$ neurons. Since $f$ is $L$-Lipschitz, one can control the uniform continuity parameter and take $\delta = \frac{\varepsilon}{L}$. Hence,

$$|G_\delta| = \left(\frac{1}{\delta_d}\right)^d = \left(\frac{L\sqrt{d}}{\varepsilon}\right)^d.$$

$\qquad\square$

### A.4 A super polynomial separation

#### A.4.1 Properties of $m$

Towards the proof of Theorem 7, we start by collecting several facts about the function $m$. The first one follows from standard coupling arguments. We omit the proof.

**Claim 8.** *$m$ is monotone.*

We now show that $m$ is smooth and Lipschitz continuous. Those are immediate consequences of the fact that $m$ is the harmonic extension of a bounded function. For completeness, below, we give more self-contained arguments.

**Lemma 9.** *Let $\mathbf{p}, \mathbf{p}' \in [0,1]^{n \times n}$. Then,*

$$|m(\mathbf{p}) - m(\mathbf{p}')| \leq n\|\mathbf{p} - \mathbf{p}'\|.$$

*In other words, $m$ is $n$-Lipschitz.*

*Proof.* Let $\mathbf{p}, \mathbf{p}'' \in [0,1]^{n \times n}$ be such that $\mathbf{p} - \mathbf{p}'' = \rho e_{ij}$, where, for $i, j \in [n]$, $e_{ij}$ is a standard basis vector in $\mathbb{R}^{n \times n}$ and $\rho > 0$. Let $\mathcal{U} := \{U_{ij}\}_{i,j=1}^n$ be *i.i.d.* random variables, uniformly distributed on $[0,1]$. We couple the random graphs, $G(\mathbf{p}), G(\mathbf{p}'')$ in the following way,

$$(i,j) \in G(\mathbf{p}) \iff U_{ij} \geq \mathbf{p}_{ij} \text{ and } (i,j) \in G(\mathbf{p}'') \iff U_{ij} \geq \mathbf{p}''_{ij}.$$

We slightly abuse notation, and write $m(G(\mathbf{p}))$ for the indicator that $G(\mathbf{p})$ contains a perfect matching (and similarly for $G(\mathbf{p}'')$). It is clear that,

$$|m(\mathbf{p}) - m(\mathbf{p}'')| = |\mathbb{E}\left[m(G(\mathbf{p})) - m(G(\mathbf{p}'')))\right]|.$$

Moreover if $G_{ij}$ is the sigma algebra generated by $\mathcal{U} \setminus \{U_{ij}\}$, we have.

$$|m(\mathbf{p}) - m(\mathbf{p}'')| = |\mathbb{E}\left[\mathbb{E}\left[m(G(\mathbf{p})) - m(G(\mathbf{p}'')))|G_{ij}\right]\right]|.$$

Since $G(\mathbf{p})$ and $G(\mathbf{p}'')$ agree on every edge other than $(i,j)$, it is readily seen that, almost surely, $\mathbb{E}\left[m(G(\mathbf{p})) - m(G(\mathbf{p}'')))|G_{ij}\right] \leq \rho$. Hence,

$$|m(\mathbf{p}) - m(\mathbf{p}'')| \leq \rho.$$

For general $\mathbf{p}, \mathbf{p}' \in [0,1]^{n \times n}$ with Cauchy-Schwartz's inequality, we conclude,

$$|m(\mathbf{p}) - m(\mathbf{p}')| \leq \sum_{i,j=1}^n |p_{ij} - p'_{ij}| \leq n\sqrt{\sum_{i,j=1}^n |p_{ij} - p'_{ij}|^2} = n\|\mathbf{p} - \mathbf{p}'\|.$$

$\square$

**Lemma 10.** *The function $m$ is a $C_\infty$ function.*

*Proof.* We sketch the proof. Enumerate all $n!$ matching of the complete bipartite graph with $A = B = n$ and denote this set by $\mathcal{M}$. For $M \in \mathcal{M}$ let $A_M$ be the event that $M$ occurs in a graph $G(\mathbf{p})$ sampled according to the input vector $\mathbf{p}$. Then the probability $G(\mathbf{p})$ has a perfect matching is $P = \Pr(\bigcup_{M \in \mathcal{M}} A_M)$. Since each edge occurs independently we have using inclusion-exclusion that $P$ is a sum of polynomials in the coordinates of $\mathbf{p}$. The claim follows. $\square$

#### A.4.2 The function $m$ cannot be approximated by a polynomial monotone network

*Proof of Lemma 6.* Suppose towards a contradiction that there is a monotone network $N$ of size $s = e^{o(\log(n)^2)}$ that approximates $m$ in $[0,1]^{n \times n}$ within error less than $\frac{1}{2}$. Then, restricting $N$ to Boolean inputs, and applying a threshold gate to the output, based on whether the output is larger than $\frac{1}{2}$ would yield a monotone threshold circuit $C_N$ that computes $m$ (exactly) on Boolean inputs and has size $s + 1$.

It is known that the monotone circuit complexity of a circuit with AND and OR gates (De Morgan circuit) computing a threshold function with positive coefficients is polynomial [4]. Therefore, we claim that, the existence of $C_N$ entails the existence of a monotone circuit $C'_N$ with AND and OR gates of size[9] $O(s^t)$, for some constant $t \geq 1$, that decides if a bipartite graph has a perfect matching.

Indeed, simply replace every positive threshold gate, $g$, in $C_N$ by a monotone De Morgan circuit $C_g$ computing $g$. As the size of $C_g$ is polynomial in the number of inputs to $g$ (and hence upper bounded by $s^k$ for an appropriate constant $k$) these replacements result in a polynomial blowup with respect to the size of $C_N$: the size of $C'_N$ is at most $(s+1)s^k = O(s^{k+1})$. Therefore setting $t$ to be $k+1$ we have that the size of $C'_n$ is at most $O(s^t) = e^{t \cdot o(\log^2 n)} = e^{o(\log^2 n)}$. Since $e^{\log(n)^2} = n^{\log(n)}$, this contradicts Theorem 5. $\qquad\square$

### A.4.3 Approximating $m$ with a general network

This subsection is devoted to the proof of Lemma 7. Recall that $A$ is a polynomial time random algorithm, that approximates $m$. Moreover, on input $\mathbf{p}$, $A$ first truncates every coordinate of $\mathbf{p}$ to the first $q(n)$ most significant bits. Therefore, if $\widetilde{\mathbf{p}}$ is the vector obtained from $\mathbf{p}$ by truncating bits, we have an equality in law,

$$A(\mathbf{p}) \overset{\text{law}}{=} A(\widetilde{\mathbf{p}}). \tag{4}$$

Keeping this in mind, we shall first require the following technical result.

**Lemma 11.** *Let $\delta \in (0,1)$ be an accuracy parameter. Then, if $\tilde{\mathbf{p}} \in [0,1]^{n \times n}$ is such that every coordinate $\tilde{\mathbf{p}}_{\mathbf{ij}}$ can be represented by at most $q(n)$ bits,*

$$\mathbb{P}\left(|m(\tilde{\mathbf{p}}) - A(\tilde{\mathbf{p}})| > \delta\right) \leq 2e^{-r(n)\delta^2/3}. \tag{5}$$

*As a consequence, for every $\mathbf{p} \in [0,1]^{n \times n}$,*

$$\mathbb{P}\left(|m(\mathbf{p}) - A(\mathbf{p})| > \delta + \frac{n^2}{\sqrt{2}^{q(n)}}\right) \leq 2e^{-r(n)\delta^2/3}.$$

*Proof.* We first establish the first part. Indeed, since $A(\tilde{\mathbf{p}})$ can be realized a linear combination of *i.i.d.* Bernoulli's and since $\mathbb{E}[A(\tilde{\mathbf{p}})] = m(\tilde{\mathbf{p}})$, (5) is a consequence of Chernoff's inequality and follows directly from the discussion in [27, Page 73]. Now, $\tilde{\mathbf{p}}$ was obtained from $\mathbf{p}$ by keeping the $q(n)$ most significant bits. Thus, $\|\mathbf{p} - \tilde{\mathbf{p}}\| \leq \sqrt{\frac{n^2}{2^{q(n)}}}$, and by Lemma 9,

$$|m(\mathbf{p}) - m(\tilde{\mathbf{p}})| \leq \frac{n^2}{\sqrt{2}^{q(n)}}.$$

So, because of (4), and (5),

$$\mathbb{P}\left(|m(\mathbf{p}) - A(\mathbf{p})| > \delta + \frac{n^2}{\sqrt{2}^{q(n)}}\right) = \mathbb{P}\left(|m(\mathbf{p}) - A(\tilde{\mathbf{p}})| > \delta + \frac{n^2}{\sqrt{2}^{q(n)}}\right)$$

$$\leq \mathbb{P}\left(|m(\mathbf{p}) - m(\tilde{\mathbf{p}})| + |m(\tilde{\mathbf{p}}) - A(\tilde{\mathbf{p}})| > \delta + \frac{n^2}{\sqrt{2}^{q(n)}}\right)$$

$$\leq \mathbb{P}\left(|m(\tilde{\mathbf{p}}) - A(\tilde{\mathbf{p}})| > \delta\right) \leq 2e^{-r(n)\delta^2/3}.$$

$\qquad\square$

We may now prove Lemma 7.

---

[9] By size we mean the number of AND as well as OR gates in the circuit.

*Proof of Lemma 7.* By standard results regarding universality of Boolean circuits [33, 37] algorithm $A$ can be implemented by a neural network $N'$ of polynomial size that get as additional input $n^2 q(n) r(n)$ random bits. In light of this, our task is to show how to get rid of these random bits. The number of points in $[0, 1]^{n \times n}$ represented by at most $q(n)$ bits in each coordinate is at most $2^{n^2 q(n)}$. On the other hand by the first part of Lemma 11 the probability for a given input $\tilde{\mathbf{p}}$ (with each coordinate in $\tilde{\mathbf{p}}$ represented by at most $q(n)$ bits) that $|N'(\tilde{\mathbf{p}}) - m(\tilde{\mathbf{p}})| \geq \delta$ is $2^{-\Omega(r(n))}$. As long as $r(n)$ is a polynomial with a large enough degree, with respect to $q(n)$, we have that $2^{n^2 q(n) - \Omega(r(n))} < 1$. Hence, there must exist a fixed choice of for the $n^2 q(n) r(n)$ (random) bits used by the algorithm such that for *every* input $\mathbf{p}$ the additive error of the neural network $N'$ on $p$ is no larger than $\delta$. Fixing these bits and hard-wiring them to $N'$ results in the desired neural network $N$. The final result follows from the proof of the second part of Lemma 11. $\qquad\square$