# OpenReview forum: "Size and depth of monotone neural networks: interpolation and approximation"
_NeurIPS.cc/2022/Conference — NeurIPS 2022 Accept_

### Official Review · Reviewer_CbVM · 2022-07-09

**Rating:** 4
**Confidence:** 4
**Soundness:** 3 good
**Presentation:** 3 good
**Contribution:** 2 fair

**Summary:**

The authors consider interpolation of monotone data (Theorem 1) and approximation of monotone functions (Theorem 2) by neural networks of 4 layers induced by the threshold activation function. The required number of neurons for the approximation is analyzed in Theorem 3.

**Questions:**

The authors should comment on how noise could be taken into consideration in their study. Is it possible to derive similar results when a continuous activation function is used?

**Limitations:**

No. It would be interesting to consider noisy data and continuous activation functions.

**Strengths And Weaknesses:**

The results in the paper verifies the role of depth in interpolation and approximation of monotone functions. These are interesting theoretical findings and help us understand the role of depth in deep learning.

However, the results cannot be sued in practical applications. The threshold activation function is not continuous, which is difficult for implementations. The monotonicity of data or multivariate functions is often ruined by noise in practice.

---

> ### Author Response · Authors · 2022-07-29
> **Rebuttal**
>
> Thank you very much for your feedback and for raising several important and interesting questions.
>
> - “These are interesting theoretical findings and help us understand the role of depth in deep learning.”
>
> **Comment**:  Thank you! Indeed, this is a theoretical paper focused on mathematical questions.
>
> - “The threshold activation function is not continuous, which is difficult for implementations.”
> - “Is it possible to derive similar results when a continuous activation function is used?”
>
> **Answer**: We agree that the threshold activation is incompatible with gradient-based algorithms. Despite this, theoretical insights about threshold neural networks can help shed light on questions related to networks with other activations. One indication comes from the recent literature about networks with threshold gates: e.g., references [25,35] in our paper. Another reason is that results regarding thresholds can often generalize to differential units such as sigmoids; see our answer below. Finally, our study introduces a scenario where a function can be approximated at constant depth but not at depth 2 (regardless of the size). We are unaware of any similar phenomenon in the literature and believe that illustrating it for the first time is of interest, even when considering a non-differentiable activation.
>
> Let us comment a bit more about continuous activations. Since we consider monotone networks, it makes sense to focus on monotone activations. The monotone activations we looked at can roughly be classified into two classes: ReLU-like activations and sigmoidal activations of sigmoidal type. From the proof of Lemma 1, we see that the convexity of ReLU (and most of its variants) turns out to be a severe hindrance to approximation in the monotone setting. As for sigmoidal activations, observe that the threshold function can be approximated arbitrarily well, almost everywhere, by sigmoids. Thus, our approximation results apply, mutatis mutandis, to sigmoid activations as well. However, to state such results would require us to introduce an approximation parameter that will depend on the separation between data points. For this reason, we chose to focus on the threshold function, which allows us to state clean theorems while capturing the intricate subtleties introduced by having monotone constraints. Please let us know if you have any other concerns concerning the activation function. We are happy to include and expand the discussion in the final version if you think it is appropriate. Furthermore, if you have any other suggestions for other types of differentiable and monotone activation functions that could be interesting to investigate, please let us know.
>
> - “The monotonicity of data or multivariate functions is often ruined by noise in practice.”
>
> **Answer**: The effect noise has on monotone data is an interesting problem. Let us note that even mild noise models can drastically change the considered setting. Indeed, any reasonable (non-monotone) noise could turn monotone functions into non-monotone, for example, if the function is constant. Since our considered architectures always produce monotone predictors, there can be no hope in approximating such noisy functions beyond the noise level, at least not in the general case. To cope, we could add extra assumptions, like bounded noise and strict monotonicity or convexity. Another option would be to relax our constraints to allow some weak non-monotonicity. Alternatively, as in isotonic regression, one could ask for the best monotone network which fits the data, but this would require restricting the size/depth of the network beforehand. As you can see, while being an exciting and relevant question, adding noise would lead to results of very different flavors and seems out of scope for the current paper. We will mention that a limitation of this work is that it considers the noiseless setting, but we also see it as a direction for future research that can build upon and expand our work. Having said that, there is also room for optimism when considering monotone interpolation in practical settings. Monotonicity may indeed be a brittle property, but nevertheless, a theoretical analysis of the noiseless setting can be of value. We observe that several papers, such as [8,11,19, 22, 30], work with monotone data sets and train monotone networks. Those references can serve as an indication that dealing with monotone data sets is not entirely hopeless.
>
> **In conclusion:**
> As you can see from the above and the other reviews, this is a highly theoretical paper aiming to understand the interplay and possible tradeoffs between depth, efficiency, and monotonicity for approximation and interpolation problems. We hope we have explained more clearly the significance and relevance of our work. We also hope you will kindly re-evaluate your score and judge the paper according to its theoretical contributions.
>
> We would be happy to address any other concerns you might have.

---

### Official Review · Reviewer_uFX9 · 2022-07-09

**Rating:** 7
**Confidence:** 3
**Soundness:** 4 excellent
**Presentation:** 3 good
**Contribution:** 4 excellent

**Summary:**

A monotonic relationship occurs when an increase (decrease) in an input value gives an increase (decrease) in the output value. Monotone ReLU neural networks preserve this structure in the data by ensuring that the network's weights are positive. There are fundamental questions regarding monotone neural networks and the manuscript investigates: (1) [universality] Can a monotone network of depth L approximate every monotone dataset? [The manuscript proves that L = 2 is not good enough, but L = 4 is.] (2) [Approximation power] Do monotone networks need to be much larger than standard networks? [The manuscript proves that there are maps such that the difference can be larger than any polynomial in d (where d is the number of input neurons).

**Questions:**

- In Theorem 2, the authors prove that if the f is L-Lipschitz then there is a nearby 4-layer neural network with O(d(Lsqrt(d)/eps)^d) neurons. This is a super-exponential blow-up in d. Do the authors believe that O(d(Lsqrt(d)/eps)^d) is necessary? Usually, one expects that as d increases the approximation power of the network increases.

- I do not understand the comment on line 125 that [8] also constructs a neural network with comparable number of neurons to Theorem 2. It sounds like it might only take O( (L/eps)^d ).

**Limitations:**

There are limited potential negative societal impact.

**Strengths And Weaknesses:**

The manuscript presents a set of compelling universal approximation theory results for monotone neural networks. I particularly enjoyed Section 5, where the authors used ideas from monotone complexity theory of Boolean functions to construct a monotone function that are much harder (in terms of number of parameters in the neural network) to approximate with monotone neural networks.

- There is quite a big gap between O(d) neurons required in Theorem 3 and the O(d(Lsqrt(d)/eps)^d) neurons in Theorem 2. Therefore, one expects that this manuscript is not the final word.

- This manuscript only describes universal approximation theory results, which reveal that monotone neural networks can be far less expressive. There is the additional problem of training that is not discussed here. Therefore, it seems that monotone neural  networks are far from becoming a mainstream neural network model.

- The problem of monotone approximation has been considered in approximation theory since the 1970s. In the multivariable setting, ideas such as: (1) Monotone tensor product regression splines (for dimension <=5), see [1], (2) Triangulation based monotone splines, e.g., see [2] and [4]. (3) Kernel regression, see [3]. While the results in this manuscript probably do not exist in this literature, it would be nice if the authors could reference the classical literature on monotone approximation

[1] G. Beliakov, Shape preserving approximation using least squares splines, Approximation Theory Appl., 16 (2000), pp. 80–98.
[2] P. Costantini and C. Manni, A local shape-preserving interpolation scheme for scattered data, Comput. Aided Geom. Des., 16 (1999), pp. 385–405.
[3] P. Hall and L. Huang, Nonparametric kernel regression subject to monotonicity constraints, Ann. Stat., 29 (2001), pp. 624–647.
[4] K. Willemans and P. Dierckx, Smoothing scattered data with a monotone Powell-Sabin spline surface, Numer. Algorithms, 12 (1996), pp. 215–232.

WRITING
The manuscript would benefit from another round of polishing. Though, I find that manuscript is clearly written. Here is a small subset of grammar mistakes that I marked while reading. (I imagine that there are many more.)

p1, line 25: "modeling monotonic relationship" should probably read "modeling the monotonic relationship"
p4, line 134: "requires much larger size" should read "requires a much larger size"
p4, line 154: "may result with negative" should read "may result in negative"
p4, line 161: "First, as using both min and max gates in the same architecture with positive parameters do not fall into the modern paradigm of an activation function." sounds better as "First, using both min and max gates in the same architecture with positive parameters does not fall into the modern paradigm of an activation function."
p5, line 175: "applies for arbitrary dimension larger than 1" should read "applies to arbitrary dimensions larger than 1"
p5, line 180: "One difference of our" should read "One difference between our"
p5, line 185: "require super polynomial" should read "require a super polynomial"
p5, line 189: "function m, that requires" should read "function m, which requires"
p5, line 206: "Each layer is composed from" sounds better as "Each layer is composed of"
p6, line 229: "positive is indeed without" should be "positive holds without"
p9, line 342: "can can" should be "can"
p9, line 344: "each gate by a" sounds better as "each gate with a"
p9, line 361: "theses copies" should read "these copies"

---

> ### Author Response · Authors · 2022-07-29
> **Rebuttal**
>
> Thank you very much for your feedback. We appreciate your many suggestions, which will improve our paper's readability.
>
> Below we address your questions and the points you've raised:
>
> - "I particularly enjoyed Section 5, where the authors used ideas from monotone complexity theory of Boolean functions to construct a monotone function that are much harder…
> to approximate with monotone neural networks"
>
> **Comment**: Thank you! This technique of reducing lower bounds for neural networks to lower bounds on circuits seems to be quite powerful. We think there are more connections to be made between complexity theory and neural networks and hope this paper will lead to more results in this direction.
>
> - "There is the additional problem of training that is not discussed here."
>
> **Answer**: We agree. The scope of this paper is expressivity questions. There are many interesting questions regarding optimizing the loss and generalization bounds for monotone neural networks. We hope our paper will lead to further study of these questions, and we will mention some of them in a future research subsection.
>
> - "It would be nice if the authors could reference the classical literature on monotone approximation"
>
> **Answer**: Thank you very much for the references. We will be sure to incorporate these references, look for more sources, and add a proper discussion.
>
> - "The manuscript would benefit from another round of polishing"
>
> **Answer**: Since submitting the paper, we have made an independent pass and made many minor changes to the style. We also plan to make another pass before submitting the final version, and we will address all comments not captured in our previous pass. Thank you for bringing these errors to our attention.
>
> - "In Theorem 2, the authors prove that if the f is L-Lipschitz then there is a nearby 4-layer neural network with O(d(Lsqrt(d)/eps)^d) neurons. This is a super-exponential blow-up in d. Do the authors believe that O(d(Lsqrt(d)/eps)^d) is necessary? Usually, one expects that as d increases the approximation power of the network increases."
>
> **Answer**: This is a terrific question. We do not know whether our obtained bound is optimal, but it seems that at least some exponential blow-up is necessary. One factor which comes into play is the diameter of the unit cube, which scales with the dimension. Thus, as the dimension increases, there is 'more space to cover. Since our monotone networks end up being piecewise constant, it seems reasonable to expect that the number of regions on which the network is constant scales by a power of d. This leads to exponential terms in d.
>
> As you note, usually, higher dimensions allow for more parameters which tend to improve approximation power when considering the number of neurons. This difference between the general and monotone setting is precisely what we have tried to uncover in our work and why we find those questions so exciting.
>
> - "I do not understand the comment on line 125 that [8] also constructs a neural network with comparable number of neurons to Theorem 2. It sounds like it might only take O( (L/eps)^d )."
>
> **Answer**: Indeed, the number of neurons will scale like a power of L/eps. The use of iterated Riemann sums allows to bypass the so-called 'curse of dimensionality, and so it ignores the sqrt(d) factor, which is the diameter of the cube. However, note that the need to re-evaluate iterated integrals for each dimension successively could make the power larger than d. We intended to highlight possible similarities in the dependence on L and epsilon. We will make sure to emphasize that the size is comparable up to dimensional factors.

---

### Official Review · Reviewer_G2N7 · 2022-07-12

**Rating:** 7
**Confidence:** 1
**Soundness:** 3 good
**Presentation:** 3 good
**Contribution:** 3 good

**Summary:**

In this work, authors study the problem of finding size and depth efficient monotone neural networks that interpolates the monotone datasets. Among two popular choice of activation functions, i.e., ReLU and threshold activation function, authors proved that there are monotone functions that cannot be approximated within arbitrary small error by monotone neural networks with ReLU activation function. Thus, in this study monotone neural networks with threshold activations are considered.

Firstly, authors showed that 2-layered monotone neural network with threshold activation cannot interpolate monotone dataset. Later on they showed that there exist 4 layered monotone neural network with threshold activation function that can interpolate monotone dataset with n points in Rd provided the size of neural network is O(nd). The direct implication of interpolation result is that 4 layered monotone neural networks can approximate arbitrary well any monotone function over [0,1]d.  This is significant improvement over best-known previous result, which states that monotone neural network with threshold activation can approximate any monotone function however depth of the approximating network will scale linearly with the dimension of the input data.

The authors further investigated on the size required to approximate any monotone function arbitrarily close by the constant depth monotone neural network with threshold activation function. The authors showed that the inductive bias of having positive parameters can lead to a super-polynomial blow-up in the number of neurons when approximating monotone functions.


**Questions:**

At surface level it appears that results might be dependent on the choice of activation function. Is it  possible to throw some light on this aspect? If in practice the threshold function is approximated by sigmoid, will the results still hold?

**Strengths And Weaknesses:**

The paper discusses on the proof that there exist 4 layered monotone neural networks with threshold activation that can interpolate the monotone dataset with n points in Rd. Its direct implication proved that there exists the constant depth monotone network with threshold activation that can approximate the monotone functions arbitrary well. This result comes with the tradeoff over the size of the network.
The paper is well written, and references are adequate.

---

> ### Author Response · Authors · 2022-07-29
> **Rebuttal**
>
> Thank you very much for your positive feedback!
>
> Regarding your question:
>
> - “At surface level it appears that results might be dependent on the choice of activation function. Is it possible to throw some light on this aspect? If in practice the threshold function is approximated by sigmoid, will the results still hold?”
>
> **Answer**: Indeed, we copy here part of our response to Reviewer 3 (CbVM).
>
> Observe that threshold functions can be approximated arbitrarily well, almost everywhere, by sigmoids. Thus, our approximation results apply, mutatis mutandis, to sigmoid activations as well. However, to state such results would require us to introduce an approximation parameter that will depend on the separation between data points. For this reason, we chose to focus on threshold activation, which allows us to state clean theorems while capturing the intricate subtleties introduced by having monotone constraints.

---

### Meta-Review · Area_Chair_o1a4 · 2022-08-24

**Recommendation:** Accept
**Confidence:** Certain

**Metareview:**

Surprisingly strong result about the expressive power of monotone networks

**Award:**

No

---

### Decision · Program_Chairs · 2022-09-14

Accept